# Investigation on the Microstructure and Mechanical Properties of the Ti-Ta Alloy with Unmelted Ta Particles by Laser Powder Bed Fusion

**DOI:** 10.3390/ma16062208

**Published:** 2023-03-09

**Authors:** Mu Gao, Dingyong He, Li Cui, Lixia Ma, Zhen Tan, Zheng Zhou, Xingye Guo

**Affiliations:** 1Faculty of Materials and Manufacturing, Beijing University of Technology, Beijing 100124, China; 2Beijing Engineering Research Center of Eco-Materials and LCA, Beijing 100124, China

**Keywords:** titanium-tantalum alloy, metallic biomaterial, laser powder bed fusion, mechanical properties

## Abstract

Titanium-tantalum (Ti-Ta) alloy has excellent biomechanical properties with high strength and low Young’s modulus, showing great application potential in the biomedical industry. In this study, Ti-Ta alloy samples were prepared by laser powder bed fusion (LPBF) technology with mixed pure 75 wt.% Ti and 25 wt.% Ta powders as the feedstock. The maximum relative density of Ti-Ta samples prepared by LPBF reached 99.9%. It is well-accepted that four nonequilibrium phases, namely, α′, α″ and metastable β phase exist in Ti-Ta alloys. The structure of α′, α″ and β are hexagonal close-packed (HCP), base-centered orthorhombic (BCO) and body-centered cubic (BCC), respectively. X-ray Diffraction (XRD) analysis showed that the α′ phase transformed to the α″ phase with the increase of energy density. The lamellar α′/α″ phases and the α″ twins were generated in the prior β phase. The microstructure and mechanical properties of the Ti-Ta alloy were optimized with different LPBF processing parameters. The samples prepared by LPBF energy density of 381 J/mm^3^ had a favorable ultimate strength (UTS) of 1076 ± 2 MPa and yield strength of 795 ± 16 MPa. The samples prepared by LPBF energy density of 76 had excellent ductility, with an elongation of 31% at fracture.

## 1. Introduction

Ti, Fe, and Co-Cr alloy, as traditional biological implant materials, have been applied for quite a long time [1,2]. The implants are prone to corrosion in the body and produce oxides or metal ions that are toxic to the human body when in long-term service [2]. The tiny particles produced from the implant part by perennial wear can lead to lesions of human cells and tissues [3]. Furthermore, their mechanical properties, such as the elastic modulus, density, and ductility of the above current commercialized materials are inadequate for biological application. The modulus of metal is typically higher than that of human bones, leading to a stress-shielding effect that is harmful to patients. The favorable metal biomaterials require good biocompatibility, excellent corrosion resistance, high strength, low elastic modulus, and lightweight [4,5] to avoid stress shielding during the service in the human body [6]. Tantalum (Ta) metal with excellent ductility, corrosion resistance, and good biocompatibility has strong resistance to grinding and deformation during service in the human body. Thus, Ta is one of the materials with the best compatibility with organisms and has great development advantages for biological implants [7,8,9]. However, it is difficult to process Ta biological devices by traditional processes due to their high melting point and high density. At the same time, it is difficult to apply Ta in large-size implants on humans [1,10]. Ti-Ta alloy, as an emerging biological material system, is gradually attracting the interest of scholars [11,12,13]. Ti and Ta are infinitely soluble with high affinity and can form a stable continuous solid solution due to their similar physical properties and cell lattice type. Compared with Ti6Al4V, Ti-Ta has similar strength but a lower modulus [6,11,13,14]. The introduction of the Ta element in the alloy leads to stronger cell adhesion and better biocompatibility, which has a positive effect on bone formation [13,15]. On the other hand, Ta can stabilize the β phase in Ti alloy, and its content also affects the mechanical properties. The microstructures of binary Ti-Ta alloys are sensitive to Ta content. The quenched alloys exhibit lamellar HCP martensite (α′) structure at a Ta content below 20%. The needle-like orthorhombic martensite (α″) structure at a Ta content in the range from 30 to 50%. The metastable β + α″ structure has a Ta content of 60%, and single metastable BCC β structure a Ta content above 60% [11]. Zhou et al. [16] studied the influence of Ta content change on the properties of the alloy, and found that the orthogonal martensite (α′’) twin structure existed in Ti-Ta alloy after solution treatment and the Ti-25Ta alloy showed the best mechanical compatibility among Ti-Ta alloys, and the Ti-Ta alloy had better prospects for biomedical applications than pure Ti or pure Ta [7,11,17,18,19].

However, the difference in melting points between Ta and Ti is large, which makes it difficult to make uniform alloying of those two elements by the traditional process. Therefore, multiple remelting processes are often required to achieve a uniform composition for the systems containing refractory metals [20,21,22]. However, this process always has complicated preparation and expensive cost. The characteristics of laser additive manufacturing (AM) technology have an instantaneous high energy source, rapid cooling environment, and layer-by-layer preparation process, which provide an effective solution for Ta and Ti alloying [23]. On the other hand, as for refractory metals, the cost of AM process is much lower than the multiple remelting processes [24]. Many researchers have studied the metallurgy technology of Ti with various refractory elements using laser powder bed fusion (LPBF) technology in recent years [25,26,27,28,29,30].

Recently, some researchers studied the technology of LPBF preparation for the Ti-Ta alloy system [17,28,29]. Sing [17] successfully produced Ti-50Ta alloy using the LPBF process by mixing elements Ti and Ta powder. The results showed that the alloy had β phase structure with a low elastic modulus (E: 75.77 ± 4.04 GPa), high strength (UTS: 924 MPa), and ductility (Elongation: 11%). Brodie [28] prepared Ti-Ta bulk material by the LPBF process. The double scanning strategy was applied to obtain a high density of the sample. Various LPBF process parameters led to different energy inputs during the preparation process. The amount of energy had a great influence on the distribution of the elements and the types of phases in samples. A large number of Ta particles were distributed in the matrix with low input energy density [27]. The unmelted Ta particles in the Ti-Ta alloy had an impact on mechanical properties, which were studied previously [28,31,32]. Zhou [11] found that microstructural heterogeneity provided a wider range of tensile behaviors for Ti-Ta alloys. In Huang’s research [32], the existence of unmelted Nb/Ta/Mo particles improved fracture toughness due to the mesostructure formation.

Most studies focused on homogeneous and uniform Ti-Ta alloy, with the goal to obtain fully melted Ta in the Ti matrix. However, the Ti-Ta alloy with fully melted Ta can be hardly obtained using LPBF technology due to the small gap between the melting point of Ta (3017 °C) and the boiling point of Ti (3287 °C). In addition, the mechanical properties of Ti-Ta alloy can be tailored by the content and distribution of the unmelted Ta. The amount of unmelted Ta particles in the matrix can be adjusted only by the process parameters of LPBF, without changing the Ta element content in Ti-Ta. However, there is limited research on the influence of unmelted Ta particles on the mechanical properties of the matrix. This study prepares the Ti-Ta alloy with the same composition by different parameters of LPBF. Different LPBF process parameters give different microstructures of Ti-Ta alloy and the content of unmelted Ta in the Ti-Ta matrix. The changes in these factors bring a significant influence on the mechanical properties of Ti-Ta alloy and provide a new design idea for the application of Ti-Ta alloy. The mechanical properties of Ti-Ta alloy can be tailored by the proper selection of the LPBF parameters according to the specific application. The unmelted Ta improves the fracture toughness, ductility, and elongation of the Ti-Ta matrix [28,32]. The effect of unmelted Ta on the mechanical properties of Ti-Ta alloy was systematically studied and discussed in this paper.

According to previous research experience, Ti-Ta alloy with an addition of 25 wt.% Ta has the best comprehensive mechanical properties. In this study, 25 wt.% Ta powder was mixed with Ti powder as the original material for research. Ti-Ta alloy with different microstructure, phase, and mechanical properties was prepared using LPBF technology with different parameters. The contents of this study were organized as follows: firstly, the preparation feasibility and relative density of the Ti-Ta sample by the LPBF were studied. Secondly, the chemical composition, phase, microstructure, and mechanical properties of Ti-Ta alloy prepared with different LPBF parameters were studied. Finally, the fracture morphology analysis was conducted to investigate the strengthening mechanism of the Ta particle on the Ti-Ta alloy. The mechanical properties and microstructure of the Ti-Ta metallic composite matrix can be controlled based on the above investigations.

## 2. Materials and Methods

### 2.1. Materials and Their Characteristics

Spherical Ta powder (15–45 µm, Suzhou JunDan New Material Science and Technology Co., Ltd., Suzhou, China) and spherical Ti powder (15–53 µm, AVI Metal Powder Metallurgy Technology Co., Ltd., Beijing, China), as listed in Table 1, were mixed according to the mass ratio of Ta: Ti = 1:3 (25 wt.% Ta, 75 wt.% Ti). The powders were mixed by a three-dimensional tumble mixer with a speed of 12 r/min, and a time of 12 h. The commercial Ti6Al4V (TC4) powder (AVI Metal Powder Metallurgy Technology Co., Ltd.) with a size range of 15–53 µm was also used as the comparison sample. The shape and distribution of Ti-Ta powders were observed by scanning electron microscopy (SEM) QUANTA 650 (FEI, Lincoln, NE, USA) using the backscattering mode (BSE). Particle size distributions of Ti powder, Ta powder, and mixed Ti-Ta powder were measured by Nano Measurer 1.2 software, and the 1052 particles were calculated. The fluidity was measured by the Hall flow meter according to ASTM B213.

### 2.2. Preparation and Optimized Processing Parameters of Ti-Ta

The metallographic samples (10 mm × 10 mm × 5 mm) were prepared using the LPBF technology to investigate the relationship between the preparation parameters and its microstructure, as shown in Figure 1. The EOS M 100 (EOS GmbH, Krailling, Germany) with a beam diameter of 40 µm was used for the LPBF preparations in this work. The preparation process was carried out by the alternative scanning strategy in each layer rotated by 67° to its precursor. The interior chamber was set in an argon atmosphere with oxygen content lower than 0.1% to prevent oxidation of the alloy. The key parameters in the SLM process included laser power (*P*), scanning speed (*v*), hatch spacing (*h*), and layer thickness (*d*). In this research, *d* was fixed as 0.02 mm. The parameters of *P*, *v*, *h* were designed by DOE and set to further investigate the effect of parameters on densification by using the response surface method [33]. The LPBF parameters of all Ti-Ta samples were listed in Table 2. Archimedes’ method was used to measure the density of LPBFed Ti-Ta samples. The samples (50 mm × 5 mm × 3 mm) were prepared with the optimized parameters for the mechanical test. The electrical discharge machining (EDM) process was used to process the stretch pattern according to ASTM B557M-15, as shown in Figure 1. Ti-6Al-4V (TC4) and pure Ta were prepared using LPBF technology as a comparison to the Ti-Ta alloy in this work. The LPBF parameter of TC4 was provided by EOS GmbH company (*P* = 100 W, *v* = 1400 mm/s, *h* = 0.06 mm, *d* = 0.02 mm). The parameters of Ta were the same as TC4 except that the v was different (*v* = 200 mm/s for pure Ta).

### 2.3. Composition Analysis and Microstructure Study

The X-Z surface of the square samples and the tensile samples (after the tensile test fracture) were processed for metallographic analysis, and the surfaces of the samples were ground with SiC sandpaper #400, #1000, #3000, and #5000 successively. Subsequently, the samples were polished by silica suspension (OPS) containing 30% hydrogen peroxide for 20 min. Ultrasonic cleaning was conducted using alcohol after polishing. Kroll reagent (4% HF, 6% of HNO3) was used to etch the Ti-Ta samples at room temperature for 60 s. The samples for the relative density test and EBSD analysis were not corroded.

The oxygen content of the mixed powder was examined by an oxygen-nitrogen-hydrogen analyzer G8 Galileo (Bruker, Berlin, Germany). The phase analysis of LPBF Ti-Ta alloy samples was carried out by X-ray diffraction (XRD). The Cu Kα (λ = 1.5405 A) was selected as the irradiation source. The tube voltage was 40 kV, the tube current was 30 mA, the scanning step was 0.02°, the scanning range was 30–90° and the scanning speed was 10°/min.

The SEM QUANTA 650 (FEI, USA) equipped with a backscattered electron detector (BSE) and energy dispersive spectrometer (EDS) was used to observe the microstructure of the LPBF Ti-Ta samples, and the composition was analyzed by EDS to verify the homogeneity of the alloy. The grain orientation, grain morphology, texture, and phase distribution of the alloy were analyzed by inverse polar pattern (IPF) generated by the EBSD technique, using SEM QUANTA 650 (FEI, USA) with a scanning step of 0.05 µm. The EBSD samples were prepared by an ion polishing instrument PECS II 685 (Gatan, Pleasanton, CA, USA). The samples were polished for 15 min at an acceleration voltage of 6 kV and an incident angle of 6°.

Transmission electron microscopy (TEM) JEM-2100 (Jeol, Tokyo, Japan) was used to further analyze the influence of different preparation processes on the phase structures. The transmission sample was first cut to the thickness of 0.3 mm, and then ground to below 70 µm with sandpaper. A precision ion polishing system PIPS 691 (Gatan) was used to subsequently polish the film sample. The acceleration voltage was 6 kV. The incident angle was 10–15°.

### 2.4. Mechanical Properties Test

A uniaxial tensile test was conducted using a tension machine CMT 4204 (MTS, Eden Prairie, MN, USA) equipped with a 20 kN load cell and extensometer. The loading direction of the tensile test was perpendicular to the building direction (load applied along the Y direction), and the strain rate was 0.001/s until fracture. Each test group included three samples for reproducibility. TC4 and pure Ta were used as the comparison group. The fracture morphology was analyzed using the SEM QUANTA 650 (Quanta, Boynton Beach, FL, USA).

## 3. Results and Discussion

### 3.1. Powder Characteristics

The particle size and oxygen content of the feedstock powders are listed in Table 3. The oxygen content of Ti-Ta powder was intermediate between pure Ta and pure Ti powders. This indicated that the powder was not over-oxidized during the mixing process.

As shown in Figure 2a,b, the bright Ta particles are evenly distributed among the dark Ti particles, and the mixed powders maintain good sphericity. The mixed Ti-Ta particle size was in the range of 7–50 µm, which was suitable for the LPBF process in this work. However, a few fine particles with a diameter of 5.6 µm were generated after mixing, as shown in Figure 2c. The particle size of Ti-Ta powder mixed in this was suitable for the LPBF process. The Ta powder had the best fluidity, as shown in Figure 2d. The fluidity of mixed Ti-Ta powder was better than that of Ti powder and commercial LPBF TC4 powder. As the result, the mixed Ti-Ta powder had a suitable particle size, good sphericity, and high fluidity, which was suitable for the LPBF process.

### 3.2. Effect of Processing Parameters on Densification

Response surface design was made in Design-Expert 12 software. According to the ANOVA analysis of variance, the fitting degree R-square was 0.9718. The effect of *P*, *v* and *h* on relative density (*RD*) was fitted by a second-order function, as shown in Equation (1). The Adj R-square and Pred R-square were 0.9356 and 0.8537, respectively. Statistical indexes indicated that the established model could accurately reflect the effect of variables on the response. When the *p*-Value of *P*, *v*, *h* was less than 0.05, which indicated that the *P*, *v*, *h* were the most significant factors [34].
*RD* = 99.16 + 0.775*P* − 2.77*v* − 1.21*h* + 0.725*Ph* + 0.3*vh* − 1.35*P*^2^ − 3.5*v*^2^ − 1.98*h*^2^(1)

The relative density map of the LPBF-built Ti-Ta alloy with various process parameters were shown in Figure 3. The curved surfaces were characterized by different degrees of density, with red color indicating high density and blue color indicating low density. The process window displaying different densities was projected at the bottom plane. For a certain *h*, as shown in Figure 3, with the scanning speed increasing, the relative density increased first and then decreased. The preferred processing window was located around the *P* for 100 W and *v* for 1100 mm/s. If *h* was 0.06 mm, the relative density reached as high as over 99.50% with a power of 90–110 W and speed of 500–1200 mm/s. The highest relative density of the Ti-Ta alloy reached 99.98% when the *P*, *v*, and *h* were 100 W, 1100 mm/s, and 0.06 mm, respectively. The energy density of this processing parameter was 76 J/mm^3^. In this research, the Ti-Ta samples with four different LPBF parameters (S1–S4) were taken for further investigation, as listed in Table 4. The Image-based method was used to analyze the formation of LPBF-built Ti-Ta, as shown in Figure 4. The relative densities of all samples under those four parameters were 98.76%, 99.67%, 99.98%, and 93.74%, respectively.

The laser energy density was inversely proportional to *v*. Under the same *P*, *h,* and *d*, the lower scanning speed caused higher energy density and vice versa [27]. The scanning speed used for S1 was the lowest among all samples. The formation of S1 (Figure 4a) was worse than both S2 (Figure 4b) and S3 (Figure 4c). The decrease in the laser scan speed led to an increase in input energy which caused the keyhole to produce extensive material vaporization [35,36,37]. There were a large number of pores generated, leading to the relative density decreasing. Very limited pores were generated in S2 and S3 samples, which indicated that the process parameters of those two samples were appropriate. The formation of S4 was the worst, as shown in Figure 4d. There were a large number of irregular pores in the sample. The scanning speed used for S4 was the highest among all samples. The input laser energy was insufficient to melt the metal powder, which led to a bad combination of the material. Thus, a great number of defects appeared in the sample. Due to the terrible formation of LPBF-built S4, it was not chosen for further investigation in this study.

The low-magnification (70×) SEM images of the samples S1, S2, and S3 were shown in Figure 5a–c. Unmelted Ta particles (white particles in the figure) were found in the matrix. The imaging method was used to make statistics of unmelted Ta particles in the matrix. The unmelted Ta particles accounted for 0.7 vol.%, 5.1 vol.%, and 10.3 vol.% in samples S1, S2, and S3 respectively. The amount of the unmelted Ta particles was increasing with the decrease of the energy density. It was obviously found that the S3 contains the most unmelted Ta particles. There were few unmelted Ta particles that existed in S1. The laser energy was much higher than S3 which led to the molten pool having a higher temperature during the LPBF process. Thus, the Ta particles tended to gain enough energy for full melting.

### 3.3. Microstructure of LPBF Ti-Ta Alloy

The uniformity of element distribution was investigated using EDS for the three Ti-Ta samples (S1–S3), as shown in Figure 6a–c. The Ta content of S1 was 30.4 wt.% which was the highest among all samples, due to the evaporation of Ti under the condition of high input energy. The Ta content of S2 and S3 were 28.3 wt.% and 27.9 wt.%, respectively, which were similar to the designed composition.

The Ta content was increased around the incompletely melted Ta particles, forming a Ta-rich zone. It had an influence on the phase species and the crystal structure in LPBF-built Ti-Ta alloy. The centers of unmelted Ta particles were taken to conduct linear EDS scanning for determining the change in Ta content. S2 and S3 were selected for testing. In Figure 7a,b, the highest Ta content was 32.8 wt.% and 36.7 wt.% in S2 and S3 respectively. The lowest Ta content was 19 wt.% and 15.4 wt.% in S2 and S3 respectively. As the line scanning results indicated, the unmelted Ta particle content was close to the design value in both S2 and S3. However, the Ta content fluctuated at different positions on the scanning line.

The input energy density of S2 was higher than that of S3 during the preparation process. Therefore, more Ta was melted, and the evaporation of Ti was high, resulting in the Ta content in the matrix of S2 being higher than that of S3. Both in S2 and S3, the Ta content in some positions was more than 30 wt.% but less than 50 wt.%, suggesting the α″ phase tended to be formed instead of the β phase [11].

### 3.4. Phase Identification of LPBF Ti-Ta Alloy

The phases and structures of the LPBF-built Ti-Ta samples were detected by the XRD, as shown in Figure 8. The α′, α″, and β phases were identified by XRD according to the PDF cards 03-065-9615 00-52-920, and 00-052-960. In Figure 8a, all the peaks of the α″ phase appeared in the XRD pattern which indicated S1 was mainly the α″ phase. The peak of β (1 1 0) overlapped with α″ (0 0 2) and β (2 1 1) did not appear, which demonstrated a small amount of β phase existed in the S1. The input energy and the temperature of the molten pool in S1 were high, which led to the intense evaporation of Ti. This decreased the content of Ti in the composition and promoted the formation of the α″ in S1. The amount of melted Ta in the Ti-Ta matrix of S1 did not reach the minimum limit to stabilize the β phase, so the amount of β phase was very low. Meanwhile, the β phase in the S1 was also attributed to a few unmelted Ta particles. In the XRD pattern of the S2, as shown in Figure 8b, all the α′, α″, and β phases were detected. The existence of the α″ phase was distinguished from the α′ phase by the occurrence of α″ (0 2 0) and α″ (0 2 1) peaks. The energy density used for preparing S2 was lower than S1, thus the evaporation of Ti was less. Hence, the average Ta content of the matrix was lower, which promoted the formation of the α′ phase. The α′ and β phases were found in the XRD pattern of the S3, as shown in Figure 8c. The unmelted Ta particles caused component heterogeneity and segregation, resulting in the formation of the Ta rich zone, and thus the β phases tend to generate. However, the Ta content in the Ti-Ta matrix in S3 was less than the minimum limit to form a β-phase Ti-Ta matrix [11], as shown in Figure 8c. Therefore, the β peaks existed because of the plenty of unmelted Ta particles in S3.

According to XRD data of S1, S2, and S3, the grain size of Ti-Ta alloy was calculated by the Debye-Scherrer formula,
(2)$$D=\frac{K\gamma }{B\cos\theta }$$
where D was the grain size, K was the constant of Scherrer, *γ* was the X-ray wavelength, usually set at 0.15406 nm as liquid, *B* was the full width at half maxima of XRD diffraction peak, and *θ* was the Bragg diffraction angle. The calculated grain sizes of S1, S2, and S3 were 16.31 nm, 10.67 nm, and 20.16 nm, respectively.

The EBSD and TEM tests of S1 and S3 were conducted to confirm the co-existence of the α′, α″, and β phases. The S1 primarily contained the α″ phase, as detected in Figure 9a, which was consistent with the XRD results. The high laser energy density caused high residual thermal stress generated in the samples during the LPBF process. The preformed under-layer was heat treated during the LPBF process which induced the formation of the annealing twin crystal due to its fast-cooling speed. In addition, the α″ phase was orthorhombic and the lattice symmetry was lower than that of HCP (α′ phase) and BCC (β phase) [17,28], which made it easy to have the twin structure.

The TEM electron diffraction pattern of the S3 (Figure 9b) showed that the S3 sample mainly contained the α′ phase. The Ta particles were detected as the β phase as shown in Figure 9b. The TEM result was consistent with the XRD results.

The S2 and S3 were selected for EBSD analysis. The selected region for the EBSD analysis did not contain the unmelted Ta particles. Figure 10a and Figure 11a were the SEM images of the S2 and S3, respectively. Both these two samples were dominated by the α′ phase, accounting for more than 90%. There were only 8% α″ phase and 2% β phase in S2. However, the content of the α″ phase was about 4% in S3, and the β phase hardly appeared.

Figure 10c and Figure 11c showed the IPF diagram of the grain orientation of S2 and S3 respectively. There was no obvious preferred orientation in S2 and S3. In S2, the grain size of α′/α″ phase was 0.2–5.14 µm and it showed granular morphology. In sample S3, the grain size of α′ phase was 0.92–7.75 µm, and it showed the columnar morphology in the building direction. This phenomenon was because of the difference in the composition supercooling due to the various LPBF energy densities. Martensite texture was significantly increased, as shown in Figure 10d and Figure 11d. The α′/α″ were formed by the solid transition of the β phase during the cooling process.

The microstructures of LPBF Ti-Ta alloy prepared with different energy densities were different. Obviously, equiaxed prior β phase was observed in the S1 sample (Figure 12a), and the martensite α′ phase with lamellar and acicular characteristics was further observed inside the prior β phase by the local magnified image (Figure 11b), which was similar to the as-cast sample [11,16]. The reason for the generation of coarser prior β phase grain was related to significant thermal interaction between interlayers, i.e., the previous as-built layer suffered a heat treatment provided by the later printed layer. The α″ phase began to precipitate within the prior β phase during subsequent solidification.

The microstructure morphology of sample S2 was similar to S1, as shown in Figure 11c. The lower LPBF energy density was applied to prepare S2. Thus, the effect of inter alternating heat was weaker than that of S1 and the degree of supercooling was increased. Grain growth was restrained during the solidification process. The lamellar martensite α′ phase existed in the prior β phase. Compared with Figure 11b,d, the α′ phase in S2 was in needle-like, which was thinner than the α″ in S1.

Some unmelted or incompletely melted Ta particles embedded in the Ti-Ta matrix were observed in Figure 12e. The obvious molten pool occurred in the morphology of S3. At high magnification (Figure 12f), the prior β phase was found to grow along the radial direction of the molten pool in form of a columnar grain. The grain boundary of the β phase remained at room temperature, and the fine lamellar martensite α′ phase precipitated inside it. The LPBF energy density of S3 was the lowest among S1, S2, and S3, and the thermal effect between layers was the weakest. Therefore, the former as-built part of the sample did not undergo further phase transformation in the subsequent LPBF printing process. Furthermore, the prior β phase did not grow up and form the large-size grain boundaries.

The grain morphology of the prior β phase is generally determined by the temperature gradient (G) and solidification rate (R) [12,27], and the critical condition for component supercooling is expressed as the following equation,
(3)$$\frac{G}{R}=-\frac{{\mathrm{mC}}_{0}\left(1-k\right)}{k}=\frac{{\Delta T}_{0}}{D}$$
where k is the partition coefficient, D is the liquid diffusivity, and ΔT_0_ is the temperature difference between the equilibrium liquidus line and solidus line of the alloy of composition C_0_. When the G/R ratio is greater than that of the ΔT_0_/D, the column grain tended to form. Otherwise, the equiaxed grain tends to form. In sample S2, firstly, although the high laser energy density led to a large degree of supercooling, the alternating heat treatment affected the as-built part of the sample during the LPBF process. Secondly, the thermal influence of each layer during the preparation process reduced the supercooling degree of the whole environment, which meant the G/R ratio was reduced. The equiaxed prior β grain was preferred to grow [27]. In S3, the applied laser energy density was low, causing the thermal effect between each as-built layer to be little. The temperature gradient was increased which led to a high degree of supercooling per layer in the LPBF process. Higher G/R led grains to grow along the epitaxy of solidification and the shape of the prior β grain tended to be columnar [27].

### 3.5. Mechanical Properties of LPBF Ti-Ta Alloy

Quasi-static uniaxial tensile tests were performed on LPBF-built Ti-Ta prepared with different energy densities. The engineering stress-strain curves and mechanical properties were displayed in Figure 13 and Table 5, respectively. Obviously, the tensile and yield strengths of LPBF TC4 were higher than those of all LPBF Ti-Ta samples. The elastic stages of the strain-stress curves of samples were scaled up as shown in Figure 13. For LPBFed Ti-Ta alloy, the minimum elongation with the value of about 1.5% was obtained in S1, which was related to the appearance of poor toughness α″ phase. In addition, S1 had no plastic stage and the fracture occurred almost at the elastic stage because of its high porosity. The initial internal defects had a serious impact on its mechanical properties.

The LPBF energy densities applied to S2 (about 381 J/mm^3^) and S3 (about 76 J/mm^3^) were lower than S1, which resulted in fewer defects and a higher relative density of Ti-Ta samples. Thus, both S2 and S3 showed an increase in ultimate tensile strength compared with conventionally wrought grade 1 CP Ti (170–458 MPa [11,38]) and comparable values to conventionally produced Ti-Ta (560 MPa [16]). In addition, the S2 was mainly dominated by α′ phase and a little of the lamellar α″ phase, which was beneficial to the increase of strength. The tensile strength of sample S2 was over 1000 MPa, which was the closest to that of TC4 and much higher than Ta. The elongation with a value of about 7.5% was significantly higher than that of the S1 sample. The sample S3 had supreme ductility with a maximum elongation of about 31%, which was contributed by the toughening effect of the unmelted Ta particles [32]. In addition, the young’s modulus of S3 was most similar to that of as-cast Ti-Ta [11] which was the lowest among all Ti-Ta samples. Compared with pure Ta, sample S3 had higher strength, similar fractural elongation and lower modulus. In addition, Young’s modulus of pure Ti and Ta was higher than that of uniform Ti-Ta alloy [10,16,39]. The mechanical performance of Ti-Ta alloy in the previous study was listed in Table 5 as a comparison.

The fracture morphology of S2 with the highest strength and S3 with the highest ductility in LPBF Ti-Ta was further analyzed by SEM. There are significant differences in fracture types of the two samples, as shown in Figure 14a,e,c,g. There was no shrinkage at the fracture of S2, showing the characteristics of mixed fracture. Inversely, there was obvious plastic deformation and shrinkage at the fracture of S3, indicating the characteristics of ductile fracture. As shown in Figure 14b,f, the cleavage planes and shallow dimples existed, indicating sample S2 was dominated by brittle fracture. The continuous deep and large dimples were discovered in the fracture of S3, indicating that the ductile fracture mechanism was dominant for sample S3. Figure 14c,g shows the SEM images of the longitudinal section of the fracture of samples S2 and S3 respectively. Obviously, both the prior β phase grain and Ta particles at the fracture of sample S2 remained equiaxed, indicating the tensile sample suffered less deformation. In addition, some voids and cracks distributed along grain boundary were also observed, further indicating that the crack propagated along the prior β phase grain boundaries. However, the unmelted Ta particles were significantly elongated along the tensile direction, indicating that the Ta particles undertook a large amount of deformation. Therefore, the high ductility of sample S3 was mainly dominated by appropriate unmelted ‘softer’ Ta particles.

## 4. Conclusions

In this study, the difference in energy density led to an influence on heat accumulation, and supercooling degree of the inner layer, resulting in the difference in composition, phase structure, and microstructure. It had a great influence on mechanical properties. The main conclusions of this study are as follows:The influences of different printing parameters on the formability of Ti-Ta alloy were investigated. The Ta particles melted sufficiently with very high energy density but will cause a large number of pores and defects. The high-density Ti-Ta alloy (>99.9%) with many unmelted Ta particles was obtained by using low LPBF energy. More defects and pores were generated with very low energy density, which insufficiently melted Ti-Ta alloy.The microstructure of LPBF in the Ti-Ta alloy was dominated by the α′ phase with very fine grains by using relatively low LPBF energy. The twin martensite α″ phase tended to generate with high laser energy. In the contrast, Ti-Ta of α′ phase and pure Ta particles of β phase were obtained in the alloy with the low LPBF energy. The thermal effect of the printing process had a great influence on the formation of phases and morphology.The tensile strength of the Ti-Ta alloy prepared by LPBF was higher than that of the as-cast sample. It was also higher than the previously reported LPBF Ti-Ta alloy. The sample prepared with 76 J/mm^3^ LPBF energy density showed a tensile strength of 771 MPa and a fracture elongation of 31%, which had outstanding ductility and moderate strength. The sample prepared with 381 J/mm^3^ showed a tensile strength of 1076 MPa, and a fracture elongation of 7.5%, which had excellent strength. The mechanical property of LPBF Ti-Ta alloy was dependent on the LPBF preparation energy. The unmelted Ta particles improved the toughness but reduced the strength. When the Ti-Ta alloy needed high strength, the LPBF laser energy was recommended to 381 J/mm^3^. When the Ti-Ta alloy needed high ductility the LPBF laser energy was recommended to 76 J/mm^3^.

## Figures and Tables

**Figure 1 materials-16-02208-f001:**
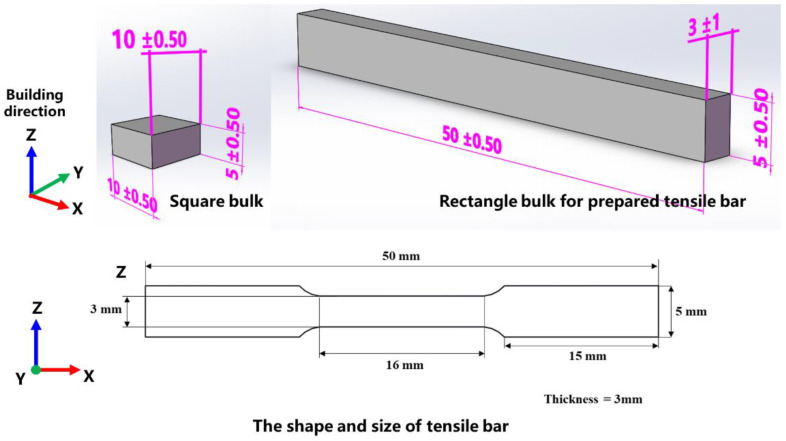
Design of bulk and tensile samples for LPBF built.

**Figure 2 materials-16-02208-f002:**
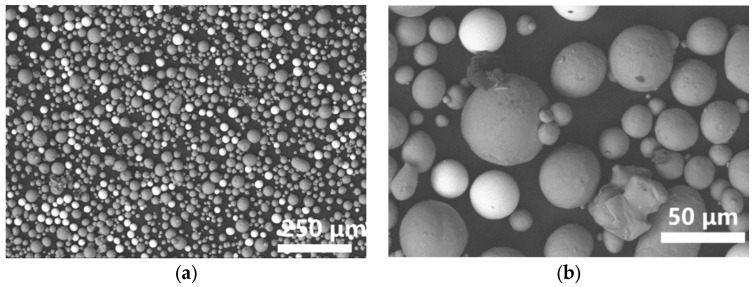
The BSE images of (**a**,**b**) mixed Ti-Ta powder, (**c**) particle size distribution of mixed Ti-Ta powders and (**d**) fluidity of original pure Ti, original pure Ta and mixed Ti-Ta powders.

**Figure 3 materials-16-02208-f003:**
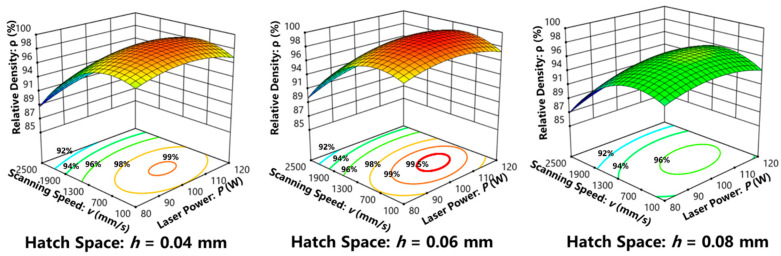
Effect of *P* and *v* on density with fixed *h*.

**Figure 4 materials-16-02208-f004:**
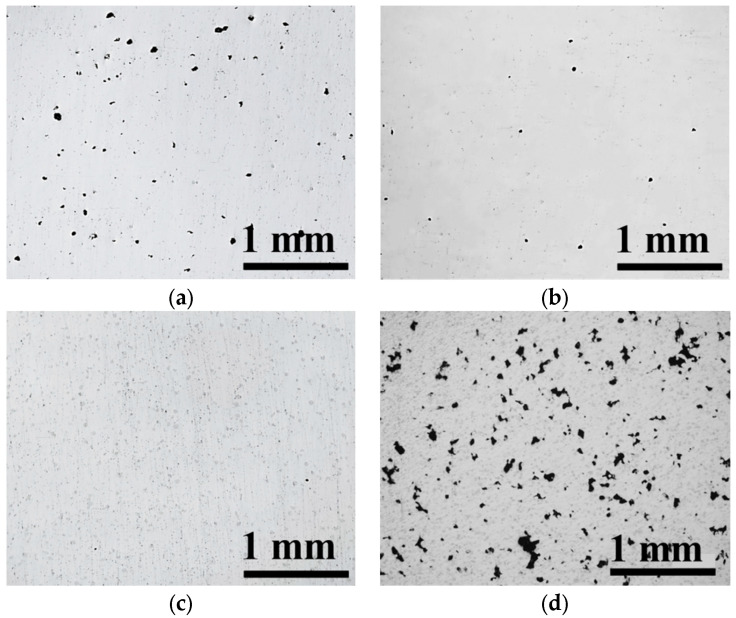
Optical micrograph of cross-sections of LPBF-built Ti-Ta obtained at the four building parameters; (**a**): S1, (**b**): S2, (**c**): S3, (**d**): S4.

**Figure 5 materials-16-02208-f005:**
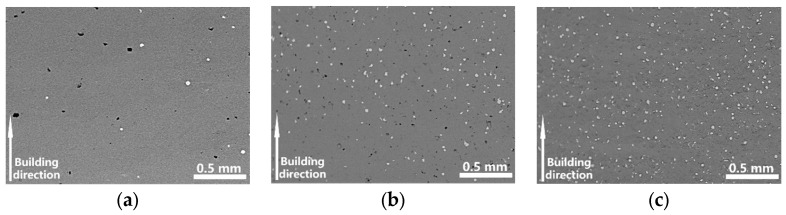
Optical micrograph of unmelted Ta particles in the etched LPBF-built Ti-Ta samples. (**a**) S1 (*v* = 120 mm/s, E = 694 J/mm^3^), (**b**) S2 (*v* = 220 mm/s, E = 381 J/mm^3^) and (**c**) S3 (*v* = 1100 mm/s, E = 76 J/mm^3^).

**Figure 6 materials-16-02208-f006:**
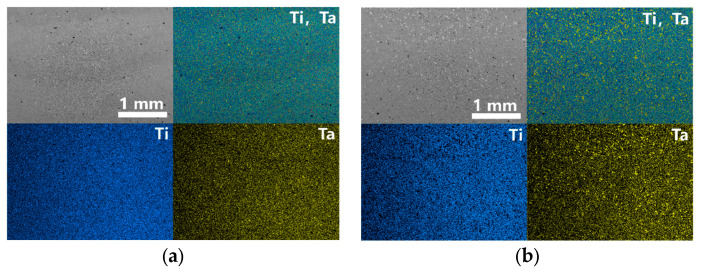
EDS surface scan results of LPBF Ti-Ta at different parameters: (**a**) sample S1 (v = 120 mm/s, E = 694 J/mm^3^), (**b**) sample S2 (v = 220 mm/s, E = 381 J/mm^3^), (**c**) sample S3 (v = 1100 mm/s, E = 76 J/mm^3^).

**Figure 7 materials-16-02208-f007:**
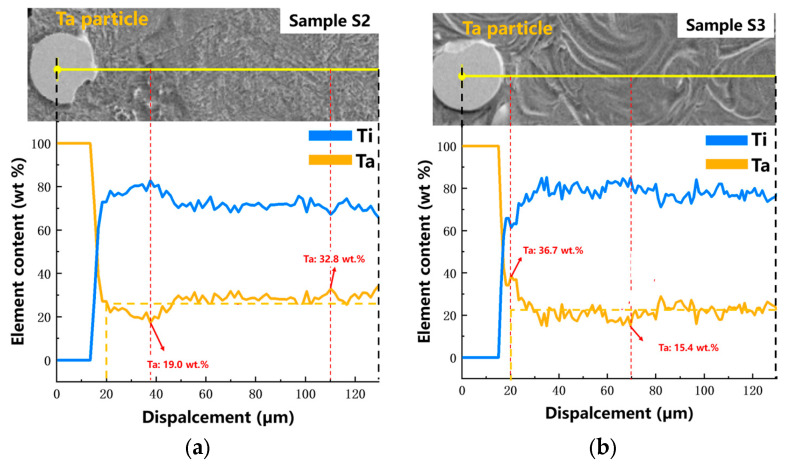
EDS linear scan results of LPBF Ti-Ta with different building parameters: (**a**) sample S2 (v = 220 mm/s, E = 381 J/mm^3^), and (**b**) sample S3 (v = 1100 mm/s, E = 76 J/mm^3^).

**Figure 8 materials-16-02208-f008:**
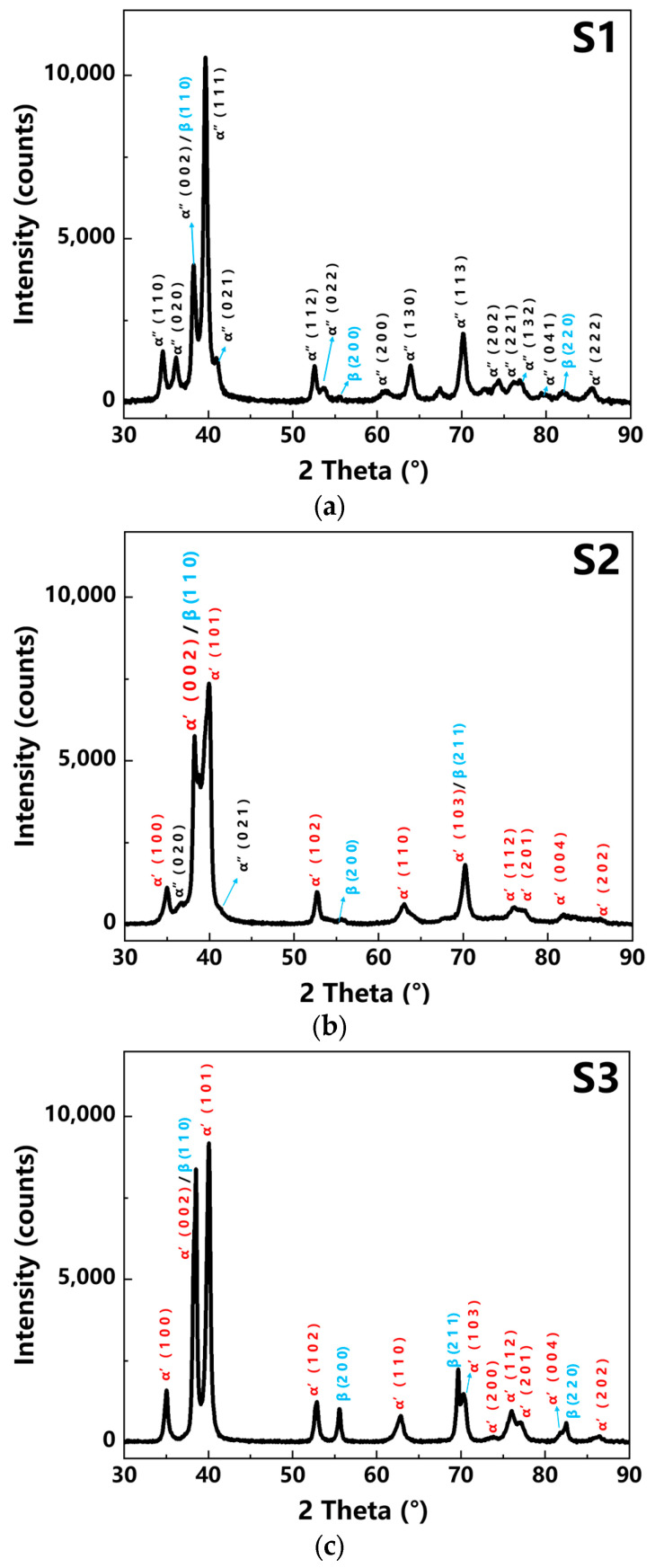
XRD pattern of LPBF Ti-Ta for different volume energy density conditions. (**a**) S1 (v = 120 mm/s, E = 694 J/mm^3^), (**b**) S2 (v = 220 mm/s, E = 381 J/mm^3^); and (**c**) S3 (v = 1100 mm/s, E = 76 J/mm^3^).

**Figure 9 materials-16-02208-f009:**
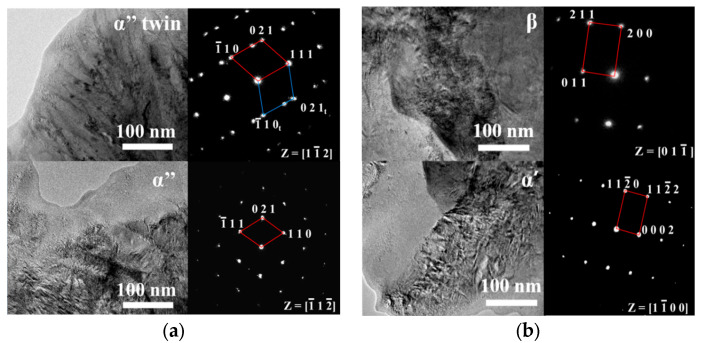
TEM bright field images of the microstructure of Ti-25Ta samples with selected area diffraction patterns. (**a**)sample S1 (v = 120 mm/s, E = 694 J/mm^3^) and (**b**) sample S3 (v = 1100 mm/s, E = 76 J/mm^3^).

**Figure 10 materials-16-02208-f010:**
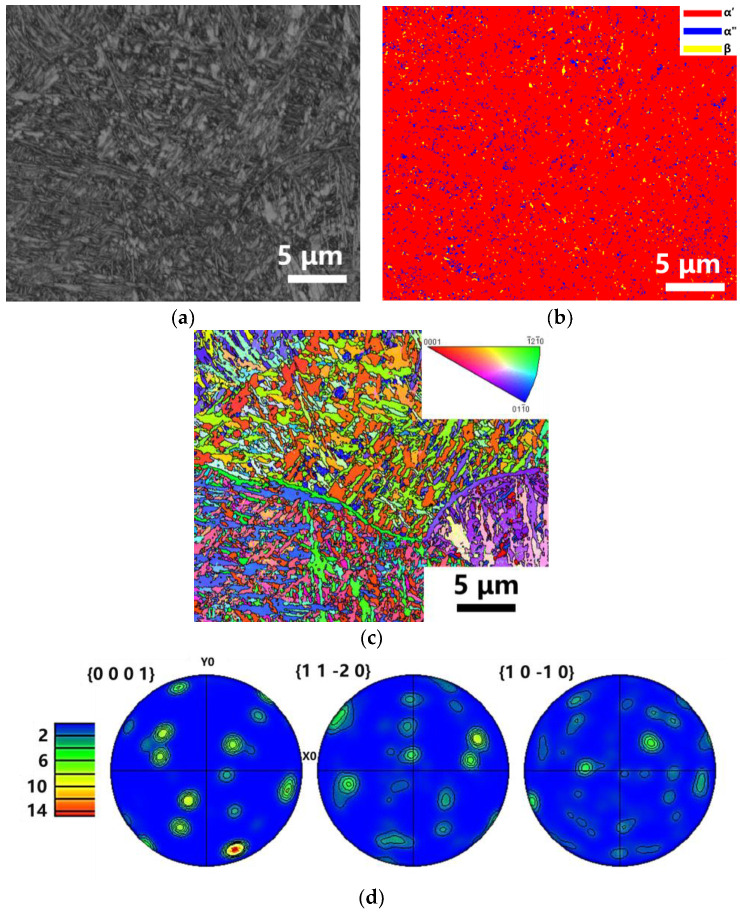
A series of images of EBSD results for sample S2 of LPBF Ti-Ta. (**a**) The position of the EBSD testing area, (**b**) The phase distribution, (**c**) IPF maps and (**d**) pole figure of α′ martensite in sample S2 (v = 220 mm/s, E = 381 J/mm^3^).

**Figure 11 materials-16-02208-f011:**
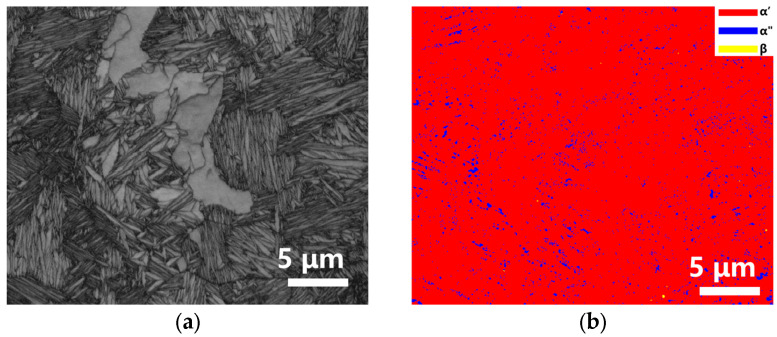
A series of images of EBSD results for sample S3 of LPBF Ti-Ta. (**a**) The position of the EBSD testing area, (**b**) The phase distribution, (**c**) IPF maps and (**d**) pole figure of α′ martensite in sample S3 (v = 1100 mm/s, E = 76 J/mm^3^).

**Figure 12 materials-16-02208-f012:**
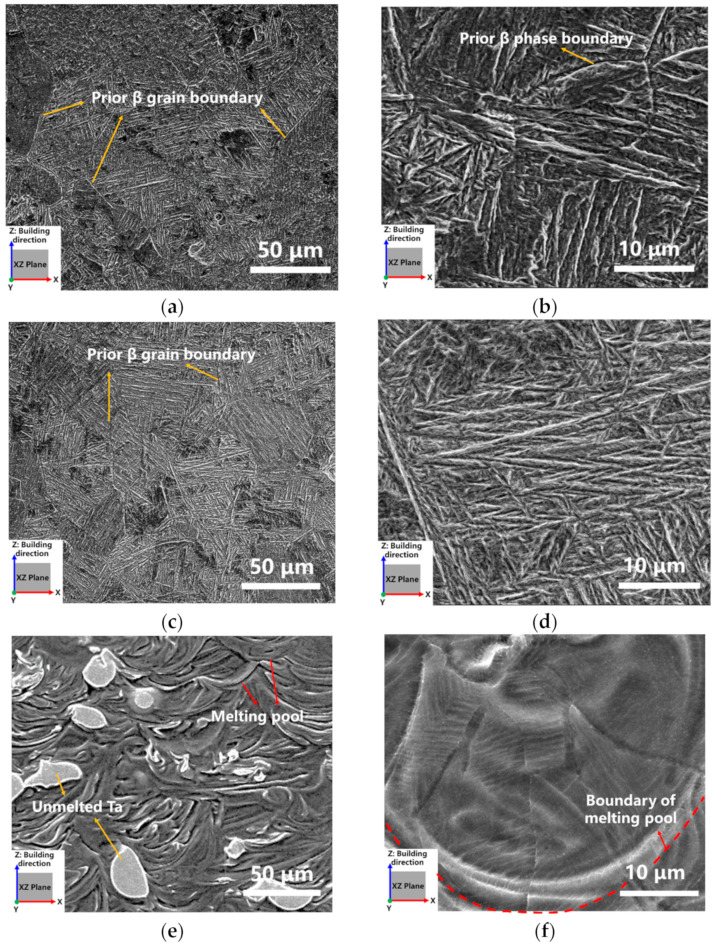
SEM images of LPBF Ti-Ta samples at various building parameters: (**a**,**b**) S1 (v = 120 mm/s, E = 694 J/mm^3^), (**c**,**d**) S2 (v = 220 mm/s, E = 381 J/mm^3^), and (**e**,**f**) S3. (v = 1100 mm/s, E = 76 J/mm^3^).

**Figure 13 materials-16-02208-f013:**
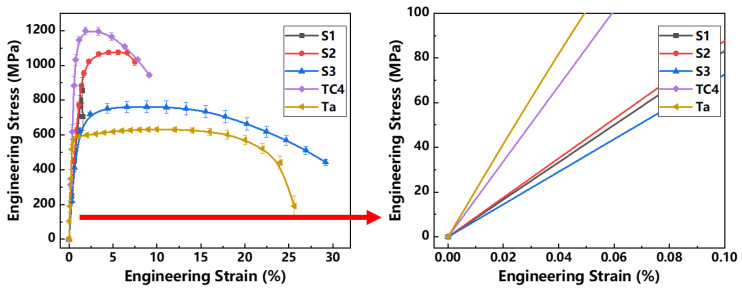
Engineering stress-strain curves of Ti-Ta samples produced by LPBF at various building parameters. The TC4 and pure Ta samples were built as a comparison to Ti-Ta.

**Figure 14 materials-16-02208-f014:**
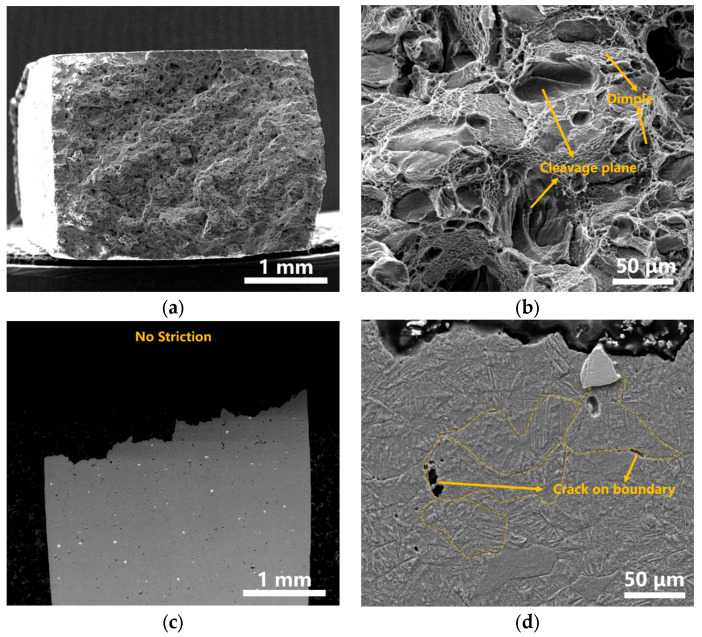
Morphology of fracture in LPBF Ti-Ta samples: (**a**,**b**) are cross-sectional fracture images of sample S2 with different magnification; (**c**,**d**) are longitudinal cross-sectional fracture images of sample S2; (**e**,**f**) are cross-sectional fracture image of sample S3 with different magnification. (**g**,**h**) longitudinal cross-sectional fracture image of sample S3.

**Table 1 materials-16-02208-t001:** The chemical composition content of pure Ti and Ta powders.

	Ti	Ta	O	Fe	N
Ti	Bal.	-	0.065 wt.%	0.059 wt.%	0.011 wt.%
Ta	-	Bal.	0.043 wt.%	0.047 wt.%	0.012 wt.%

**Table 2 materials-16-02208-t002:** The LPBF parameters of Ti-Ta.

Process Parameters	
Laser power: *P* (W)	80–120
Scan speed: *v* (mm/s)	100–2500
Hatch space: *h* (mm)	0.04–0.08
Layer thickness: *d* (mm)	0.02

**Table 3 materials-16-02208-t003:** The particle size and oxygen content of Ti, Ta and mixed Ti-Ta powders.

Type of Powder	Particle Size (µm)	Oxygen Content (ppm)
Ta	19.1–47.7	430
Ti	7.4–50.8	650
Mixed Ti-Ta	5.6–49.4	590

**Table 4 materials-16-02208-t004:** The LPBF parameters of Ti-Ta samples for investigated.

Sample Name	Laser Power *P* (W)	Scan Speed *v* (mm/s)	Hatch Space *h* (mm)	Layer Thickness *d* (mm)	Energy Density *ρ* (J/mm^3^)
S1	100	120	0.06	0.02	691
S2	100	220	0.06	0.02	381
S3	100	1100	0.06	0.02	76
S4	100	2500	0.06	0.02	33

**Table 5 materials-16-02208-t005:** The mechanical properties of Ti-Ta pure Ta and TC4 by LPBF.

Sample Name	Yield Strength (MPa)	Tensile Strength (MPa)	Fracture Strain (%)	Young’s Modulus (GPa)
S1	693 ± 52	856 ± 113	1.5 ± 0.2	75 ± 1
S2	795 ± 16	1076 ± 2	7.5 ± 0.4	80 ± 2
S3	589 ± 17	771 ± 30	31.0 ± 1.0	65 ± 2
Ta	575 ± 16	632 ± 11	26.0 ± 2.6	187 ± 4
TC4	1106 ± 10	1204 ± 15	9.1 ± 0.3	148 ± 1
Ti-25Ta [11,16] (Arc melting)	480	575	20	64
Ti-25Ta [28] (LPBF)	486	540	18	65
Ti-30Ta [11,16] (Arc melting)	441	740	15.2	73
Ti-50Ta [11,16] (Arc melting)	378	609	18.3	83
Ti-50Ta [17] (LPBF)	883	925	11.7	76
Ti-70Ta [11,16] (Arc melting)	374	580	6	67

## Data Availability

Data from this study will be available upon request.

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
