# Peer review of "Investigation on the Microstructure and Mechanical Properties of the Ti-Ta Alloy with Unmelted Ta Particles by Laser Powder Bed Fusion"

_materials, 2023, doi:10.3390/ma16062208_

Round 1
Reviewer 1 Report
1- The introduction section needs to be updated a little more.
2- Calculating particle sizes from XRD data using the Debye–Scherrer formula further improves the article.
3- Comparison with previous studies makes the article even better.
4- English should be improved.
Author Response
Q1- The introduction section needs to be updated a little more.
Response: Thank you for your valuable suggestions. We modified the Introduction section and emphasized the novelty of the study. The modified parts are marked in red colour in page 2-3.
In page 2, line 50-54: we added an explanation of the structure of α' α" and β phase: “The microstructures of binary Ti-Ta alloys are sensitive to Ta content. The quenched alloys exhibit lamellar HCP martensite (α') structure at a Ta content below 20%. The needle-like orthorhombic martensite (α") structure at a Ta content in the range from 30 to 50%. The metastable β + α" structure at a Ta content of 60%, and single metastable BCC β structure at a Ta content above 60% [11].”
In page 2-3, line 85-103: we modified the description of the novelty about this study: “Most studies focused on homogeneous and uniform Ti-Ta alloy, whose goal was to obtain the fully melted Ta in the Ti matrix. However, the Ti-Ta alloy with the fully melted Ta can be hardly obtained using LPBF technology due to the small gap between the melting point of Ta (3017℃) and the boiling point of Ti (3287°C). In addition, the mechanical properties of Ti-Ta alloy can be tailored by the content and distribution of the unmelted Ta. The amount of unmelted Ta particles in the matrix can be adjusted only by the process parameters of LPBF, without changing the Ta element content in Ti-Ta. However, there is limited research on the influence of unmelted Ta particles on the mechanical properties of the matrix. This study prepares the Ti-Ta alloy with the same composition by different parameters of LPBF. Different LPBF process parameters give different microstructures of Ti-Ta alloy and the content of unmelted Ta in the Ti-Ta matrix. The changes in these fac-tors bring a significant influence on the mechanical properties of Ti-Ta alloy and provide a new design idea for the application of Ti-Ta alloy. The mechanical properties of Ti-Ta alloy can be tailored by the proper selection of the LPBF parameters according to the specific application. The unmelted Ta improves the fracture toughness, ductility and elongation of Ti-Ta matrix [28, 32]. The effect of unmelted Ta on the mechanical properties of Ti-Ta alloy was systematically studied and discussed in this paper.”
Q2- Calculating particle sizes from XRD data using the Debye–Scherrer formula further improves the article.
Response: Thank you for your valuable comments and suggestions. We added the part of Debye–Scherrer formula for calculating the grain size of three sample, the new part was marked in red color.
In page 12, Table 352-358: “According to XRD data of S1, S2 and S3, the grain size of Ti-Ta alloy was calculated by Debye-Scherrer formula,
(2)
where D was the grain size, K was the constant of Scherrer, γ was the X-ray wavelength, usually set 0.15406 nm as liquid, B was the full width at half maxi-ma of XRD diffraction peak, and θ was the Bragg diffraction angle. The calculated grain size of S1, S2 and S3 were 16.31 nm, 10.67 nm and 20.16 nm, respective-ly.”
Q3- Comparison with previous studies makes the article even better.
Response: Thank you for your valuable comments and suggestions. We listed some result of mechanical properties of Ti-Ta alloy from other scholars’ pervious study in Table 5. The LPBF-built Ti-Ta in this study exhibited an excellent mechanical property. According to the requirement of application, the LPBF parameters can be adjusted to prepare Ti-Ta alloy with different performance index. In page 18, Table 5:
Table 5. The mechanical properties of Ti-Ta pure Ta and TC4 by LPBF
|
Sample name |
Yield strength (MPa) |
Tensile strength (MPa) |
Fracture strain (%) |
Young's modulus (GPa) |
|
S1 |
693 ± 52 |
856 ± 113 |
1.5 ± 0.2 |
75 ± 1 |
|
S2 |
795 ± 16 |
1076 ± 2 |
7.5 ± 0.4 |
80 ± 2 |
|
S3 |
589 ± 17 |
771 ± 30 |
31.0 ± 1.0 |
65 ± 2 |
|
Ta |
575 ± 16 |
632 ± 11 |
26.0 ± 2.6 |
187 ± 4 |
|
TC4 |
1106 ± 10 |
1204 ± 15 |
9.1 ± 0.3 |
148 ± 1 |
|
Ti-25Ta [11, 16] (Arc melting) |
480 |
575 |
20 |
64 |
|
Ti-25Ta [28] (LPBF) |
486 |
540 |
18 |
65 |
|
Ti-30Ta [11, 16] (Arc melting) |
441 |
740 |
15.2 |
73 |
|
Ti-50Ta [11, 16] (Arc melting) |
378 |
609 |
18.3 |
83 |
|
Ti-50Ta [17] (LPBF) |
883 |
925 |
11.7 |
76 |
|
Ti-70Ta [11, 16] (Arc melting) |
374 |
580 |
6 |
67 |
Q4- English should be improved.
Response: Thank you for your valuable suggestions. We checked the figures and tables to make it looks nice and concise. In addition, we reviewed the whole manuscript and polish the language.

Reviewer 2 Report
The manuscript entitled: "Investigation on the Microstructure and Mechanical Properties of the Ti-Ta Alloy with Unmelted Ta Particles by Laser Powder Bed Fusion" by Mu Gao, Dingyong He, Li Cui, Lixia Ma, Zhen Tan, Zheng Zhou, Xingye Guo can be published with minor corrections.
1. The manuscript is quite well written, but it would be helpful for the reader to explain what the (alpha’), (alpha’’) and (beta) phases mean (in the abstract and in later sections).
2. The novelty of the study should be explained more strongly (in the Introduction section).
3. Figures 2, 7, and 8 vary in quality (Figure 8 versus Figures 2 and 7).
Author Response
Q1- The manuscript is quite well written, but it would be helpful for the reader to explain what the a(alpha’), a(alpha’’), and b (beta) phases mean (in the abstract and in later sections).
Response: Thank you for your valuable comments and suggestions. We added an explanation of the structure of α', α", and β phase in the abstract and introduction.
In page 1, abstract, line 14-17: we added explanation of the mean of α' α" and β phase: “It is well accepted that four nonequilibrium phases, namely, α', α" and metastable β phase exist in Ti-Ta alloys. The structure of α', α" and β are hexagonal close-packed (HCP), base centered orthorhombic (BCO) and body centered cubic (BCC), respectively.”
In page 2, introduction section, line 50-54: we presented the formation of α' α" and β phase in Ti-Ta alloy with different Ta content: “The microstructures of binary Ti-Ta alloys are sensitive to Ta content. The quenched alloys exhibit lamellar HCP martensite (α') structure at a Ta content below 20%. The needle-like orthorhombic martensite (α") structure at a Ta content in the range from 30 to 50%. The metastable β + α" structure at a Ta content of 60%, and single metastable BCC β structure at a Ta content above 60% [11]..”
Q2- The novelty of the study should be explained more strongly (in the Introduction section).
Response: Thank you for your valuable comments and suggestions. In page 2-3, line 85-103: we modified the description of the novelty about this study: “Most studies focused on homogeneous and uniform Ti-Ta alloy, whose goal was to obtain the fully melted Ta in the Ti matrix. However, the Ti-Ta alloy with the fully melted Ta can be hardly obtained using LPBF technology due to the small gap between the melt-ing point of Ta (3017℃) and the boiling point of Ti (3287°C). In addition, the mechanical properties of Ti-Ta alloy can be tailored by the content and distribution of the unmelted Ta. The amount of unmelted Ta particles in the matrix can be adjusted only by the process parameters of LPBF, without changing the Ta element content in Ti-Ta. However, there is limited research on the influence of unmelted Ta particles on the mechanical properties of the matrix. This study prepares the Ti-Ta alloy with the same composition by different parameters of LPBF. Different LPBF process parameters give different microstructures of Ti-Ta alloy and the content of unmelted Ta in the Ti-Ta matrix. The changes in these factors bring a significant influence on the mechanical properties of Ti-Ta alloy and provide a new design idea for the application of Ti-Ta alloy. The mechanical properties of Ti-Ta alloy can be tailored by the proper selection of the LPBF parameters according to the specific application. The unmelted Ta improves the fracture toughness, ductility and elongation of Ti-Ta matrix [28, 32]. The effect of unmelted Ta on the mechanical properties of Ti-Ta alloy was systematically studied and discussed in this paper.”
Q3. Figures 2, 7, and 8 vary in quality (Figure 8 versus Figures 2 and 7).
Response: Thank you for your valuable comments and suggestions. We remake the XRD pattern curves in Fig. 8. The old figures were substituted by the new figures.
